# Tracking Time-varying Graphical Structure

**Erich Kummerfeld**
Carnegie Mellon University
Pittsburgh, PA 15213
ekummerf@andrew.cmu.edu

**David Danks**
Carnegie Mellon University
Pittsburgh, PA 15213
ddanks@andrew.cmu.edu

## Abstract

Structure learning algorithms for graphical models have focused almost exclusively on stable environments in which the underlying generative process does not change; that is, they assume that the generating model is globally stationary. In real-world environments, however, such changes often occur without warning or signal. Real-world data often come from generating models that are only locally stationary. In this paper, we present *LoSST*, a novel, heuristic structure learning algorithm that tracks changes in graphical model structure or parameters in a dynamic, real-time manner. We show by simulation that the algorithm performs comparably to batch-mode learning when the generating graphical structure is globally stationary, and significantly better when it is only locally stationary.

## 1    Introduction

Graphical models are used in a wide variety of domains, both to provide compact representations of probability distributions for rapid, efficient inference, and also to represent complex causal structures. Almost all standard algorithms for learning graphical model structure [9, 10, 12, 3] assume that the underlying generating structure does not change over the course of data collection, and so the data are i.i.d. (or can be transformed into i.i.d. data). In the real world, however, generating structures often change and it can be critical to quickly detect the structure change and then learn the new one.

In many of these real-world contexts, we also do not have the luxury of collecting large amounts of data and then retrospectively determining when (if ever) the structure changed. That is, we cannot learn in "batch mode," but must instead learn the novel structure in an online manner, processing the data as it arrives. Current online learning algorithms can detect and handle changes in the learning environment, but none are capable of general, graphical model structure learning.

In this paper, we develop a heuristic algorithm that fills this gap: it assumes only that our data are *locally* i.i.d., and learns graphical model structure in an online fashion. In the next section, we quickly survey related methods and show that they are individually insufficient for this task. We then present the details of our algorithm and provide simulation evidence that it can successfully learn graphical model structure in an online manner. Importantly, when there is a stable generating structure, the algorithm's performance is indistinguishable from a standard batch-mode structure learning algorithm. Thus, using this algorithm incurs no additional costs in "normal" structure learning situations.

## 2    Related work

We focus here on graphical models based on directed acyclic graphs (DAGs) over random variables with corresponding quantitative components, whether Bayesian networks or recursive Structural Equation Models (SEMs) [3, 12, 10]. All of our observations in this paper, as well as the core

algorithm, are readily adaptable to learn structure for models based on undirected graphs, such as Markov random fields or Gaussian graphical models [6, 9].

Standard graphical model structure learning algorithms divide into two rough types. Bayesian/score-based methods aim to find the model $M$ that maximizes $P(M|Data)$, but in practice, score the models using a decomposable measure based on $P(Data|M)$ and the number of parameters in $M$ [3]. Constraint-based structure learning algorithms leverage the fact that every graphical model predicts a pattern of (conditional) independencies over the variables, though multiple models can predict the same pattern. Those algorithms (e.g., [10, 12]) find the set of graphical models that best predict the (conditional) independencies in the data.

Both types of structure learning algorithms assume that the data come from a single generating structure, and so neither is directly usable for learning when structure change is possible. They learn from the sufficient statistics, but neither has any mechanism for detecting change, responding to it, or learning the new structure. Bayesian learning algorithms—or various approximations to them—are often used for online learning, but precisely because case-by-case Bayesian updating yields the same output as batch-mode processing (assuming the data are i.i.d.). Since we are focused on situations in which the underlying structure can change, we do not *want* the same output.

One could instead look to online learning methods that track some environmental feature. The classic TDL algorithm, TD(0) [13], provides a dynamic estimate $E_t(X)$ of a univariate random variable $X$ using a simple update rule: $E_{t+1}(X) \leftarrow E_t(X) + \alpha(X_t - E_t(X))$, where $X_t$ is the value of $X$ at time $t$. The static $\alpha$ parameter encodes the learning rate, and trades off convergence rate and robustness to noise (in stable environments). In general, TDL methods are good at tracking slow-moving environmental changes, but perform suboptimally during times of either high stability or dramatic change, such as when the generating model structure abruptly changes.

Both Bayesian [1] and frequentist [4] online changepoint detection (CPD) algorithms are effective at detecting abrupt changes, but do so by storing substantial portions of the input data. For example, a Bayesian CPD [1] outputs the probability of a changepoint having occurred $r$ timesteps ago, and so the algorithm must store more than $r$ datapoints. Furthermore, CPD algorithms assume a model of the environment that has only abrupt changes separated by periods of stability. Environments that evolve slowly but continuously will have their time-series discretized in seemingly arbitrary fashion, or not at all.

Two previous papers have aimed to learn time-indexed graph structures from time-series data, though both require full datasets as input, so cannot function in real-time [14, 11]. Talih and Hengartner (2005) take an ordered data set and divide it into a fixed number of (possibly empty) data intervals, each with an associated undirected graph that differs by one edge from its neighbors. In contrast with our work, they focus on a particular type of graph structure change (single edge addition or deletion), operate solely in "batch mode," and use undirected graphs instead of directed acyclic graph models. Siracusa and Fisher III (2009) uses a Bayesian approach to find the posterior uncertainty over the possible directed edges at different points in a time-series. Our approach differs by using frequentist methods instead of Bayesian ones (since we would otherwise need to maintain a probability distribution over the superexponential number of graphical models), and by being able to operate in real-time on an incoming data stream.

## 3    Locally Stationary Structure Tracker (LoSST) Algorithm

Given a set of continuous variables $\mathbf{V}$, we assume that there is, at each time $r$, a true underlying generative model $\mathbf{G}^r$ over $\mathbf{V}$. $\mathbf{G}^r$ is assumed to be a recursive Structural Equation Model (SEM): a pair $\langle G, \mathbf{F} \rangle$, where $G$ denotes a DAG over $\mathbf{V}$, and $\mathbf{F}$ is a set of linear equations of the form $V_i = \sum_{V_j \in pa(V_i)} a_{ji} \cdot V_j + \epsilon_i$, where $pa(V_i)$ denotes the variables $V_j \in G$ such that $V_j \rightarrow V_i$, and the $\epsilon_i$ are normally distributed noise/error terms. In contrast to previous work on structure learning, we assume only that the generating process is *locally* stationary: for each time $r$, data are generated i.i.d. from $\mathbf{G}^r$, but it is not necessarily the case that $\mathbf{G}^r = \mathbf{G}^s$ for $r \neq s$. Notice that $\mathbf{G}^r$ can change in both structure (i.e., adding, removing, or reorienting edges) and parameters (i.e., changes in $a_{ji}$'s or the $\epsilon_i$ distributions).

At a high level, the *Lo*cally *S*tationary *S*tructure *T*racker (LoSST) algorithm takes, at each timestep $r$, a new datapoint as input and outputs a graphical model $\mathbf{M}^r$. Obviously, a single datapoint is

insufficient to learn graphical model structure. The LoSST algorithm instead tracks the locally stationary sufficient statistics—for recursive SEMs, the means, covariances, and sample size—in an online fashion, and then dynamically (re)learns the graphical model structure as appropriate. The LoSST algorithm processes each datapoint only once, and so LoSST can also function as a single-pass, graphical model structure learner for very large datasets.

Let $\mathbf{X}^r$ be the $r$-th multivariate datapoint and let $X_i^r$ be the value of $V_i$ for that datapoint. To track the potentially changing generating structure, the datapoints must potentially be differentially weighted. In particular, datapoints should be weighted more heavily after a change occurs. Let $a_r \in (0, \infty)$ be the weight on $\mathbf{X}^r$, and let $b_r = \sum_{k=1}^{r} a_k$ be the sum of those weights over time.

The weighted mean of $V_i$ after datapoint $r$ is $\mu_i^r = \sum_{k=1}^{r} \frac{a_k}{b_r} X_i^k$, which can be computed in an online fashion using the update equation:

$$\mu_i^{r+1} = \frac{b_r}{b_{r+1}} \mu_i^r + \frac{a_{r+1}}{b_{r+1}} X_i^{r+1} \tag{1}$$

The (weighted) covariance between $V_i$ and $V_j$ after datapoint $r$ is provably equal to $\mathbf{C}_{V_i,V_j}^r = \sum_{k=1}^{r} \frac{a_k}{b_r}(X_i^r - \mu_i^r)(X_j^r - \mu_j^r)$. Let $\delta_i = \mu_i^{r+1} - \mu_i^r = \frac{a_{r+1}}{b_{r+1}}(X_i^{r+1} - \mu_i^r)$. The update equation for $\mathbf{C}^{r+1}$ can be written (after some algebra) as:

$$\mathbf{C}_{X_i,X_j}^{r+1} = \frac{1}{b_{r+1}}[b_r \mathbf{C}_{X_i,X_j}^r + b_r \delta_i \delta_j + a_{r+1}(X_i^{r+1} - \mu_i^{r+1})(X_j^{r+1} - \mu_j^{r+1})] \tag{2}$$

If $a_k = c$ for all $k$ and some constant $c > 0$, then the estimated covariance matrix is identical to the batch-mode estimated covariance matrix. If $a_r = \alpha b_r$, then the learning is the same as if one uses TD(0) learning for each covariance with a learning rate of $\alpha$.

The sample size $S^r$ is more complicated, since datapoints are weighted differently and so the "effective" sample size can differ from the actual sample size (though it should always be less-than-or-equal). Because $\mathbf{X}^{r+1}$ comes from the current generating structure, it should always contribute 1 to the effective sample size. In addition, $\mathbf{X}^{r+1}$ is weighted $\frac{a_{r+1}}{a_r}$ more than $\mathbf{X}^r$. If we adjust the natural sample size update equation to satisfy these two constraints, then the update equation becomes:

$$S^{r+1} = \frac{a_r}{a_{r+1}} S^r + 1 \tag{3}$$

If $a_{r+1} \geq a_r$ for all $r$ (as in the method we use below), then $S^{r+1} \leq S^r + 1$. If $a_{r+1} = a_r$ for all $r$, then $S^r = r$; that is, if the datapoint weights are constant, then $S^r$ is the true sample size.

Sufficient statistics tracking—$\boldsymbol{\mu}^{r+1}$, $\mathbf{C}^{r+1}$, and $S^{r+1}$—thus requires remembering only their previous values and $b_r$, assuming that $a_{r+1}$ can be efficiently computed. The $a_{r+1}$ weights are based on the "fit" between the current estimated covariance matrix and the input data: poor fit implies that a change in the underlying generating structure is more likely. For multivariate Gaussian data, the "fit" between $\mathbf{X}^{r+1}$ and the current estimated covariance matrix $\mathbf{C}^r$ is given by the Mahalanobis distance $D_{r+1}$ [8]: $D_{r+1} = (\mathbf{X}^{r+1} - \boldsymbol{\mu}^r)(\mathbf{C}^r)^{-1}(\mathbf{X}^{r+1} - \boldsymbol{\mu}^r)^T$.

A large Mahalanobis distance (i.e., poor fit) for some datapoint could indicate simply an outlier; inferring that the underlying generating structure has changed requires large Mahalanobis distances over multiple datapoints. The likelihood of the (weighted) sequence of $D_r$'s is analytically intractable, and so we cannot use the $D_r$ values directly. We instead base the $a_{r+1}$ weights on the (weighted) pooled $p$-value of the individual $p$-values for the Mahalanobis distance of each datapoint.

The Mahalanobis distance of a $V$-dimensional datapoint from a covariance matrix estimated from a sample of size $N$ is distributed as Hotelling's $T^2$ with parameters $p = V$ and $m = N - 1$. The $p$-value for the Mahalanobis distance $D_{r+1}$ is thus: $p_{r+1} = T^2(x > D_{r+1}|p = N, m = S^r - 1)$ where $S^r$ is the effective sample size. Let $\Phi(x, y)$ be the cdf of a Gaussian with mean 0 and variance $y$ evaluated at $x$. Then Liptak's method for weighted pooling of the individual $p$-values [7] gives the following definition:[1] $\rho_{r+1} = \Phi(\sum_{i=1}^{r} a_i \Phi^{-1}(p_i, 1), \sqrt{\sum a_i^2}) = \Phi(\eta_{r+1}, \sqrt{\tau_{r+1}})$, where the update equations for $\eta$ and $\tau$ are $\eta_{r+1} = \eta_r + a_r \Phi^{-1}(p_r, 1)$ and $\tau_{r+1} = \tau_r + a_r^2$.

There are many ways to convert the pooled $p$-value $\rho_{r+1}$ into a weight $a_{r+1}$. We use the strategy: if $\rho_{r+1}$ is greater than some threshold $T$ (i.e., the data sequence is sufficiently likely given the current model), then keep the weight constant; if $\rho_{r+1}$ is less that $T$, then increase $a_{r+1}$ linearly and inversely to $\rho_{r+1}$ up to a maximum of $\gamma a_r$ at $\rho_{r+1} = 0$. Mathematically, this transformation is:

$$a_{r+1} = a_r \cdot max\left\{1, \frac{\gamma T - \gamma \rho_{r+1} + \rho_{r+1}}{T}\right\} \tag{4}$$

Efficient computation of $a_{r+1}$ thus only requires additionally tracking $\rho_r$, $\eta_r$, and $\tau_r$.

We can efficiently track the relevant sufficient statistics in an online fashion, and so the only remaining step is to learn the corresponding graphical model. The implementation in this paper uses the PC algorithm [12], a standard constraint-based structure learning algorithm. A range of alternative structure learning algorithms could be used instead, depending on the assumptions one is able to make.

Learning graphical model structure is computationally expensive [2] and so one should balance the accuracy of the current model against the computational cost of relearning. More precisely, graph[2] relearning should be most frequent after an inferred underlying change, though there should be a non-zero chance of relearning even when the structure appears to be relatively stable (since the structure could be slowly drifting).

In practice, the LoSST algorithm probabilistically relearns based on the inverse[3] of $\rho_r$: the probability of relearning at time $r + 1$ is a noisy-OR gate with the probability of relearning at time $r$, and a weighted $(1 - \rho_{r+1})$. Mathematically, $P_{r+1}(relearn) = P_r(relearn) + \nu(1 - \rho_{r+1}) - P_r(relearn)\nu(1 - \rho_{r+1})$, where $\nu \in [0, 1]$ modifies the frequency of graph relearning: large values result in more frequent relearning and small values result in fewer. If a relearning event is triggered at datapoint $r$, then a new graphical model structure and parameters are learned, and $P_r(relearn)$ is set to 0. In general, $\rho_r$ is lower when changepoints are detected, so $P_r(relearn)$ will increase more quickly around changepoints, and graph relearning will become more frequent. During times of stability, $\rho_r$ will be comparatively large, resulting in a slower increase of $P_r(relearn)$ and thus less frequent graph relearning.

## 3.1 Convergence vs. diligence in LoSST

LoSST is capable of exhibiting different long-run properties, depending on its parameters. *Convergence* is a standard desideratum: if there is a stable structure in the limit, then the algorithm's output should stabilize on that structure. In contexts in which the true structure can change, another desirable property for learning algorithms is *diligence*: if the generating structure has a change of given size (that manifests in the data), then the algorithm should detect and respond to that change within a fixed number of datapoints (regardless of the amount of previous data). Both diligence and convergence are desirable methodological virtues, but they are provably incompatible: *no learning algorithm can be both diligent and convergent* [5]. Intuitively, they are incompatible because they must respond differently to improbable datapoints: convergent algorithms must tolerate them (since such data occur with probability 1 in the infinite limit), while diligent algorithms must regard them as signals that the structure has changed.

If $\gamma = 1$, then LoSST is a convergent algorithm, since it follows that $a_{r+1} = a_r$ for all $r$ (which is a sufficient condition for convergence). For $\gamma > 1$, the behavior of LoSST depends on $T$. If $T < 0$, then we again have $a_{r+1} = a_r$ for all $r$, and so LoSST is convergent. LoSST is also provably convergent if $T$ is time-indexed such that $T_r = f(S_r)$ for some $f$ with $(0, 1]$ range, where $\sum_{i=1}^{\infty} (1 - f(i))$ converges.[4]

In contrast, if $T > 1$ and $\gamma > 1$, then LoSST is provably diligent.[5] We conjecture that there are sequences of time-indexed $T_r < 1$ that will also yield diligent versions of LoSST, analogously to the condition given above for convergence.

Interestingly, if $\gamma > 1$ and $0 < T < 1$, then LoSST is neither convergent nor diligent, but rather strikes a balance between the desiderata. In particular, these versions (a) tend to converge towards stable structures, but provably do not actually converge since they remain sensitive to outliers; and (b) respond quickly to change in generating structure, but only exponentially fast in the number of previous datapoints, rather than within a fixed interval. The full behavior of LoSST in this parameter regime, including the extent and sensitivity of trade-offs, is an open question for future research. For the simulations below, unsystematic investigation led to $T = 0.05$ and $\gamma = 3$, which seemed to appropriately trade off convergence vs. diligence in that context.

## 4    Simulation results

We used synthetic data to evaluate the performance of LoSST given known ground truth. All simulations used scenarios in which either the ground truth parameters or ground truth graph (and parameters) changed during the course of data collection. Before the first changepoint, there should be no significant difference between LoSST and a standard batch-mode learner, since those datapoints are globally i.i.d. Performance on these datapoints thus provides information about the performance cost (if any) of online learning using LoSST, relative to traditional algorithms. After a changepoint, one is interested both in the absolute performance of LoSST (i.e., can it track the changes?) and in its performance relative to a standard batch-mode algorithm (i.e., what performance *gain* does it provide?). We used the PC algorithm [12] as our baseline batch-mode learning algorithm; we conjecture that any other standard graphical model structure learning algorithm would perform similarly, given the graphs and sample sizes in our simulations.

In order to directly compare the performance of LoSST and PC, we imposed a fixed "graph relearning" schedule[6] on LoSST. The first set of simulations used datasets with 2000 datapoints, where the SEM graph and parameters both changed after the first 1000 datapoints. 500 datasets were generated for each of a range of $\langle \#variables, MaxDegree \rangle$ pairs,[7] where each dataset used two different, randomly generated SEMs of the specified size and degree.

Figures 1(a-c) show the mean edge addition, removal, and orientation errors (respectively) by LoSST as a function of time, and Figures 1(d-f) show the means of $\#errors_{PC} - \#errors_{LoSST}$ for each error type (i.e., higher numbers imply LoSST outperforms PC). In all Figures, each $\langle variable, degree \rangle$ pair is a distinct line. As expected, LoSST was basically indistinguishable from PC for the first 1000 datapoints; the lines in Figures 1(d-f) for that interval are all essentially zero. After the underlying generating model changes, however, there are significant differences. The PC algorithm performs quite poorly because the full dataset is essentially a mixture from two different distributions which induces a large number of spurious associations. In contrast, the LoSST algorithm finds large Mahalanobis distances for those datapoints, which lead to higher weights, which lead it to learn (approximately) the new underlying graphical model. In practice, LoSST typically stabilized on a new model by roughly 250 datapoints after the changepoint.

The second set of simulations was identical to the first (500 runs each for various pairs of variable number and edge degree), except that the graph was held constant throughout and only the SEM parameters changed after 1000 datapoints. Figures 2(a-c) and 2(d-f) report, for these simulations, the same measures as Figures 1(a-c) and 1(d-f). Again, LoSST and PC performed basically identically for the first 1000 datapoints. Performance after the parameter change did not follow quite the same pattern as before, however. LoSST again does much better on edge addition and orientation errors, but performed significantly worse on edge removal errors for the first 200 points following the

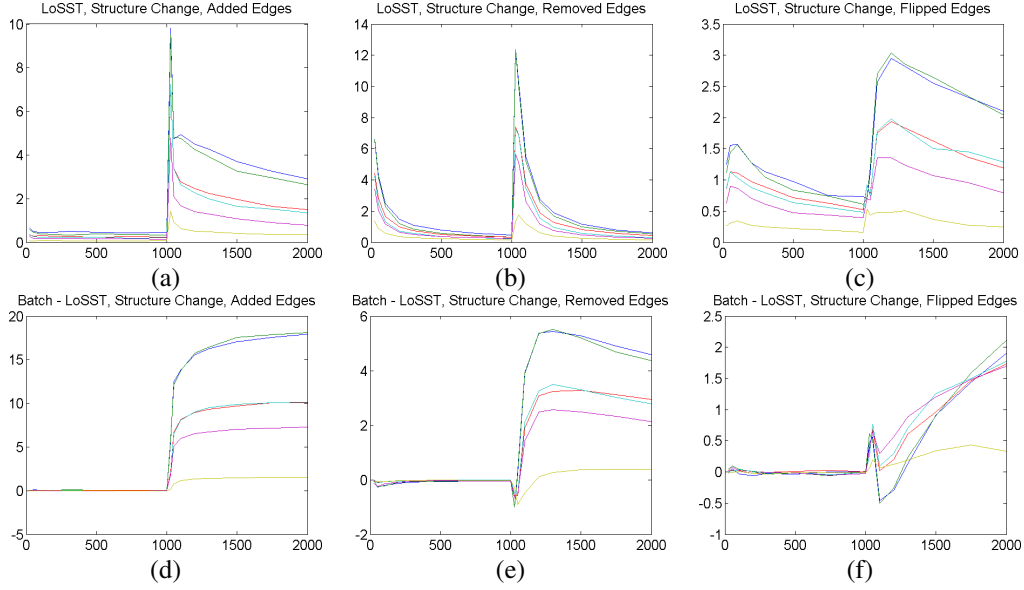

Figure 1: Structure & parameter changes: (a-c) LoSST errors; (d-f) LoSST improvement over PC

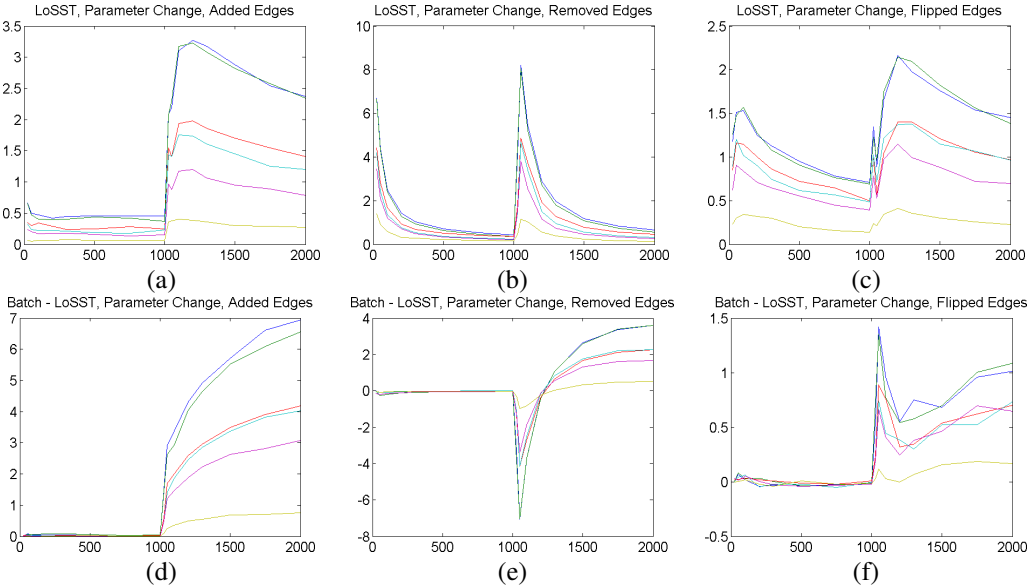

Figure 2: Parameter changes: (a-c) LoSST errors; (d-f) LoSST improvement over PC

change. When a change occcurs, PC intially responds by adding edges to the output, while LoSST responds by being more cautious in its inferences (since the effective sample size shrinks after a change). The short-term impact on each algorithm is thus: PC's output tends to be a superset of the original edges, while LoSST outputs fewer edges. As a result, PC can outperform LoSST for a brief time on the edge *removal* metric in these types of cases in which the change involves only parameters, not graph structure.

The third set of simulations was designed to explore in detail the performance with probabilistic relearning. We randomly generated a single dataset with 10,000 datapoints, where the underlying SEM graph and parameters changed after every 1000 datapoints. Each SEM had 10 variables and maximum degree of 7. We then ran LoSST with probabilistic relearning ($\nu = .005$) 500 times on this dataset. Figure 3(a) shows the (observed) expected number of "relearnings" in each 25-

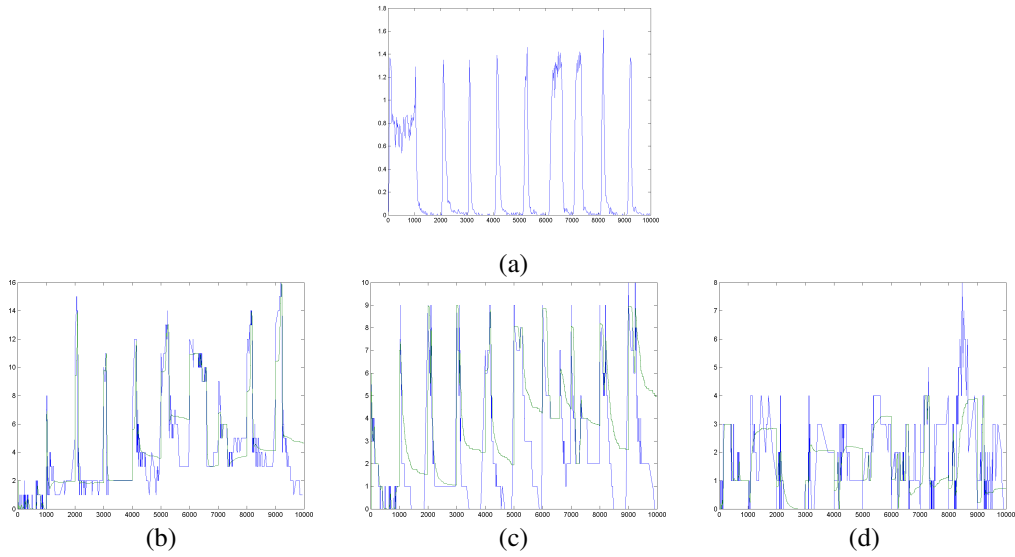

(a)

(b)                    (c)                    (d)

Figure 3: (a) LoSST expected relearnings; (b-d) Expected edge additions, removals, and flips, against constant relearning

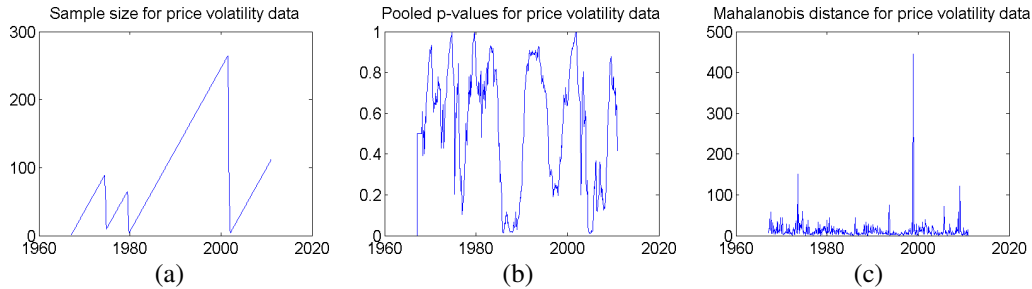

(a)                    (b)                    (c)

Figure 4: (a) Effective sample size during LoSST run on BLS data; (b) Pooled $p$-values; (c) Mahalanobis distances

datapoint window. As expected, there are substantial relearning peaks after each structure shift, and the expected number of relearnings persisted at roughly 0.1 per 25 datapoints throughout the stable periods. Figures 3(b-d) provide error information: the smooth green lines indicate the mean edge addition, removal, and orientation errors (respectively) during learning, and the blocky blue lines indicate the LoSST errors if graph relearning occurred after every datapoint. Although there are many fewer graph relearnings with the probabilistic schedule, overall errors did not significantly increase.

## 5   Application to US price index volatility

To test the performance of the LoSST algorithm on real-world data, we applied it to seasonally adjusted price index data from the U.S. Bureau of Labor Statistics. We limited the data to commodities/services with data going back to at least 1967, resulting in a data set of 6 variables: Apparel, Food, Housing, Medical, Other, and Transportation. The data were collected monthly from 1967-2011, resulting in 529 data points. Because of significant trends in the indices over time, we used month-to-month differences.

Figure 4(a) shows the change in effective sample size, where the key observation is that change detection prompts significant drops in the effective sample size. Figures 4(b) and 4(c) show the pooled $p$-value and Mahalanobis distance for each month, which are the drivers of sample size

changes. The Great Moderation was a well-known macroeconomic phenomenon between 1980 and 2007 in which the U.S. financial market underwent a slow but steady reduction in volatility. LoSST appears to detect exactly such a shift in the volatility of the relationships between these price indexes, though it detects another shift shortly after 2000.[8] This real-world case study also demonstrates the importance of using pooled $p$-values, as that is why LoSST does not respond to the single-month spike in Mahalanobis distance in 1995, but does respond to the extended sequence of slightly above average Mahalanobis distances around 1980.

# 6 Discussion and future research

The LoSST algorithm is suitable for locally stationary structures, but there are obviously limits. In particular, it will perform poorly if the generating structure changes very rapidly, or if the datapoints are a random-order mixture from multiple structures. An important future research direction is to characterize and then improve LoSST's performance on more rapidly varying structures. Various heuristic aspects of LoSST could also potentially be replaced by more normative procedures, though as noted earlier, many will not work without substantial revision (e.g., obvious Bayesian methods).

This algorithm can also be extended to have the current learned model influence the $a_r$ weights. Suppose particular graphical edges or adjacencies have not changed over a long period of time, or have been stable over multiple relearnings. In that case, one might plausibly conclude that those connections are less likely to change, and so much greater error should be required to relearn those connections. In practice, this extension would require the $a_r$ weights to vary across $\langle V_i, V_j \rangle$ pairs, which significantly complicates the mathematics and memory requirements of the sufficient statistic tracking. It is an open question whether the (presumably) improved tracking would compensate for the additional computational and memory cost in particular domains.

We have focused on SEMs, but there are many other types of graphical models; for example, Bayesian networks have the same graph-type but are defined over discrete variables with conditional probability tables. Tracking the sufficient statistics for Bayes net structure learning is substantially more costly, and we are currently investigating ways to learn the necessary information in a tractable, online fashion. Similarly, our graph learning relies on constraint-based structure learning since the relevant scores in score-based methods (such as [3]) do not decompose in a manner that is suitable for online learning. We are thus investigating alternative scores, as well as heuristic approximations to principled score-based search.

There are many real-world contexts in which batch-mode structure learning is either infeasible or inappropriate. In particular, the real world frequently involves dynamically varying structures that our algorithms must track over time. The online structure learning algorithm presented here has great potential to perform well in a range of challenging contexts, and at little cost in "traditional" settings.

**Acknowledgments**

Thanks to Joe Ramsey and Rob Tillman for help with the simulations, and three anonymous reviewers for helpful comments. DD was partially supported by a James S. McDonnell Foundation Scholar Award.

## Footnotes

[1]$\rho_{r+1}$ cannot include $p_{r+1}$ without being circular: $p_{r+1}$ would have to be appropriately weighted by $a_{r+1}$, but that weight depends on $\rho_{r+1}$.

[2]Recall that the sufficient statistics are updated after every datapoint.

[3]Recall that $\rho_r$ is a pooled $p$-value, so low values indicate unlikely data.

[4]**Proof sketch:** $\sum_{i=r}^{\infty} (1 - q_i)$ can be shown to be an upper bound on the probability that $(1 - \rho_i) > q_i$ will occur for some $i$ in $[r, \infty)$, where $q_i$ is the $i$-th element of the sequence $Q$ of lower threshold values. Any sequence $Q$ s.t. $\sum_{i=1}^{\infty} (1 - q_i) < 1$ will then guarantee that an infinite amount of unbiased data will be accumulated in the infinite limit. This provides probability 1 convergence for LoSST, since the structure learning method has probability 1 convergence in the limit. If $Q$ is prepended with arbitrary strictly positive threshold values, the first element of $Q$ will still be reached infinitely many times with probability 1 in the infinite limit, and so LoSST will still converge with probability 1, even using these expanded sequences.

[5]**Proof sketch:** By equation (4), $T > 1$ & $\gamma > 1 \Rightarrow \gamma - \frac{\gamma-1}{T} > 1 \Rightarrow a_{r+1} \geq a_r(\gamma - \frac{\gamma-1}{T}) > a_r$ for all $r$. This last strict inequality implies that the effective sample size has a finite upper bound ($= \frac{\gamma T - \gamma + 1}{(\gamma-1)(T-1)}$ if $\rho_r = 1$ for all $r$), and the majority of the effective sample comes from recent data points. These two conditions are jointly sufficient for diligence.

[6]LoSST relearned graphs and PC was rerun after datapoints $\{25, 50, 100, 200, 300, 500, 750, 1000, 1025, 1050, 1100, 1200, 1300, 1500, 1750, 2000\}$.

[7]Specifically, $\langle 4, 3 \rangle$, $\langle 8, 3 \rangle$, $\langle 10, 3 \rangle$, $\langle 10, 7 \rangle$, $\langle 15, 4 \rangle$, $\langle 15, 9 \rangle$, $\langle 20, 5 \rangle$, and $\langle 20, 12 \rangle$

[8]This shift is almost certainly due to the U.S. recession that occurred in March to November of that year.

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
