[Reviews · NeurIPS 2013]

Submitted by Assigned_Reviewer_10

Reaction to the author feedback:

I cannot agree with point (3): The work of Siracusa III and Fisher (AISTATS 2009) does *not* assume the data are iid, and it allows the "generating stucture" vary (even if only in a small set of graphical models). While the present proposed method is, in some sense, even more flexible, I'd find it misleading to claim that the proposed method is the first to address the problem.


***

The paper presents a hypothesis-testing based approach to learn graphical models in an online fashion from time series data. The proposed method is specifically targeted to scenarios where the "data generating" graphical model may change relatively frequently. The performance of the method is demonstrated using both simulated and real data.


Quality

The results seem correct. However, I find the description of related work (in Section 2) somewhat misleading: Typical score-based *algorithms* for learning Bayesian networks do not assume that the data come from a single generating model -- rather they assume the scoring function factorizes into local terms. It is true that typically studied *models* assume exchangeability (or, iid data), but this concerns the model, not the algorithm.

Clarity

While the presentation of the paper is mostly fine, there is room for improvement. Most importantly, the paper is somewhat unclear on what the actual learning task in question is: inferring change points (under some well formulated assumptions of the existence and nature of change points) or inferring the generating structure of the very last data point seen, or something else? Also, it is hard to tell whether the work is proposing a (new) model (meaning the formulations of assumptions and goals of learning) or (new) algorithm (meaning computational means to achieve the learning goals) or both.

The paragraph at the end of page 3 is difficult to follow; e.g., does the given equation for \rho_{r+1} define it or has the definition been given before? I wonder why the authors do not use periods "." when ending a sentence with a displayed equation. An almost equally important issue is the font size in Figures 1-4: could it be doubled? Also, I would not mind having descriptions for the different colors in Figures 1 and 2. In Section 6, I got confused by the idea that improved tracking could compensate for the additional computational cost -- doesn't that depend on some very domain specific valuation of tracking accuracy against compuationally cost?

Originality

The work looks original. Yet it seems to lack any real innovation or novel idea. Maybe there is something in the construction of the p-values (that I am unfortunately not able to follow completely).

Significance

The potential significance of the present work (if published) is in that it would quickly lead to the development of superior, presumably Bayesian, methods for the problem. That should indeed be viewed rather as development as opposed to research.
Summary: A heuristic-looking approach to a somewhat unclearly defined learning problem that existing machine learning machinery should already have solved.

Submitted by Assigned_Reviewer_11

Paper Summary:

The paper tackles an interesting and under explored problem of tracking changes in the underlying dependency structure between variables in a stream (e.g. time based) of data points.
The author focus on a efficient (computational/memory wise) solution for the simpler SEM models. They develop a new algorithm for this task, named LoSST. LoSST uses a standard batch learning algorithm for the underlying structure learning (PC by Spirtes et al, 2000) and tracks several statistics from the stream of data to decide whether to initiate new structure learning. The tracked statistics involve the distance from the weighted mean of each variable, the correlation matrix (eq. 1,2) and the weighted sample size. For the sample weight there is a weighting scheme for each new point based on how "surprising" the new data point is (Eq. 3,4). Each point's Mahalanobis distance (MD) is computed and the estimated p-values for the MDs by a Hotelling's T2 distribution is combined using Liptak's weighted pooling of p-values to compute an overall "significance" statistic \rho. This stat is then used in a noisy OR configuration with the previous step to decide wether to start a new structure learning. The authors compare to vanilla PC batch learning on synthetic data sets where either a change in params or a change of structure is introduced after a fixed set of datapoints, testing for different number of variables and max degree configurations. Finally, they perform a short analysis of real life data from US price index with 529 data points (every month, starting at 1967) and six variables. They show the algorithm detects some known changes/trends in the US economy while ignore short lived changes.

Pros:
1. The paper tackles an interesting and under explored problem. As the author point out there is a lot of room for additional development of practical solutions for these and other models, as well as matching theory and theoretical guarantees.
2. The algorithm is novel and practical/efficient. It makes creative use of a host of different methods for this.
3. Experiments seem adequate trying to track the effect of the various params and include an interesting real life data set.
4. Authors give a good review about previous work on this problem.
5. Overall well written.

Cons:

1. The biggest lack in the paper with respect to this interesting problem is that of theory or any guarantees under some assumptions. Convergence and diligence are like opposing forces as the authors point out (p. 4) but it seems like some form of guarantees under simplifying conditions should be plausible. At least mention as another direction for future work.
2. Similarly, the solution suggested by the authors, LoSST, is heavy on the heuristic/hack side. That's OK and enables the authors to get a practical solution, but the decisions involved should be more readably noted. Since tracking the mean and correlation matrix is straightforward, these involve mostly decisions on reweighing, effective sample size, and when/how to re learn. The authors make creative use of some methods under conditions that probably void guarantees of these methods. For example, they estimate the MD p-value using Hotelling's T2 distribution. They set the sample size N to be the effective sample size S^r. However, that distribution was developed for complete counts from a given distribution, not for partial/re weighted counts. If they actually sampled from these nets/structures with these partial weights would the distribution follow this form? Similarly, Liptak's weighted p-values was developed for meta analysis using independent tests with different sample sizes. At least if the methods are used outside their guarantees/context the authors should clearly state that, so readers don't get the wrong impression/understanding.

Other comments:

- The authors make creative use of footnotes to outline proofs, probably beyond what Leslie Lamport or the conference organizers intended. Consider using supp for that and other missing details.
- Figures are way too small, with unnecessary space between them, missing labels, missing details in the captions, legends etc. Description of the experiments is not clear enough (e.g. top paragraph p. 7 is cryptic).
- The explanation why LoSST does worse on edge addition when only the params are changed is not satisfactory.
- With regards to the above, it seems desirable that sparking params update and structure updates should be decoupled.
Summary: Really interesting direction for future research with a practical though heuristic implementation as a first stab at it.

Submitted by Assigned_Reviewer_12

This manuscript presents a systematic idea to make structure learning for SEMs
work with data sets that are only locally i.i.d., that is, the data might be
generated from distinct underlying models as time goes by. The paper is well
written and easy to follow, and addresses an important problem, namely learning
from time-dependent data, which has been neglected by many of the most used
algorithms for structure learning. The techniques to extend the learning to
time-dependent data are simple yet effective, according to the experiments with
the PC method.

There are two suggestions that could clarify the paper contributions. Firstly,
in the experiments, it is said that "any other standard graphical model
structure learning algorithms would perform similarly, ..." I believe this
assertion to be too strong. The differences among methods for structure learning
might be considerable. It is better to rephrase it, making it clearer that the
experiments are about the PC method for time-independent and with an adaptation
for time-dependent data, which is very relevant and etc (and then to say that it
is speculated that it would be the same with other methods); or to keep the
assertion and then explain in detail why that would be the case.

Secondly, in the conclusions it is discussed that applying the same ideas to
learning the structure of Bayesian networks might be not so simple of a task,
because of difficulties to keep the sufficient statistics. Even if that problem
is not necessarily central to the proposal in this paper, it shows that the use
of the same learning procedure for Bayesian networks and other models might
require more than a few tricks. Hence, I think the paper would benefit from
disclosing this fact from the beginning, maybe presenting the paper in more
direct terms about its goal into SEM models, still pointing out that the
extension to other models might be possible (but are not necessarily
straightforward). In short, the paper could be a bit sharper about its scope.
Summary: This is a well-written paper about learning the graph structure for SEM under
the assumption that data are only locally i.i.d. The formulation is simple to
understand and to implement, and experimental results look promising, even if
focused just on a comparison using the PC method.

I acknowledge to have read the feedback from the authors.
Author Feedback

Author rebuttal: We thank the reviewers for their many helpful comments and criticisms. We apologize to all for our figures being poorly formatted and labeled, and as a result, difficult to read.

Reviewer 10: (1) We agree that locality is a key feature of all score-based learning algorithms. The main point that we were attempting to make is that locality is essentially always used for the data, not just the variables. As a result, those score-based algorithms do not allow for dependencies between the scores for different data points, which is required when we have only locally i.i.d. data.

(2) In our project, the learning goal is to infer the generating structure of the most recent data point. This requires (implicitly) accurately detecting and quickly responding to change points, but that is not the explicit target of learning.

(3) We agree that Bayesian methods might be able to outperform our algorithm, though as noted in the paper, we have some concerns about the suitability of "standard" Bayesian methods when the data are only locally i.i.d. However, we think that it is notable that our algorithm is (to our knowledge) the first to address this challenging learning problem, and so provides both a demonstration that the problem is soluble, as well as a baseline against which to judge the performance of future algorithms.

Reviewer 11: Regarding performance guarantees under different assumptions, Section 3.1 shows that LoSST converges in the limit given (i) certain parametric constraints (modifying T over time in an appropriate way); (ii) the assumptions stated at the beginning of Section 3; and (iii) the existence of a stable limit model. Establishing performance guarantees when LoSST is neither convergent nor diligent (i.e. when \gamma > 1 and 0 < T < 1) is an important direction for future work.

Reviewer 12: We agree that the scope of the paper should have better explained; hopefully the responses here have helped to clarify it.